# Quantitative measurement of the histological features of alpha-1 antitrypsin deficiency-associated liver disease in biopsy specimens

**George Marek**[1]*, **Amy Collinsworth**[2], **Chen Liu**[3], **Mark Brantly**[1], **Virginia Clark**[4]

**1** Division of Pulmonary, Critical Care, and Sleep Medicine, University of Florida, Gainesville, Florida, United States of America, **2** Advanced Pathology Solutions, Little Rock, Arkansas, United States of America, **3** Department of Pathology, Yale School of Medicine, New Haven, Connecticut, United States of America, **4** Division of Gastroenterology, Hepatology, and Nutrition, University of Florida, Gainesville, Florida, United States of America

* George.Marek@medicine.ufl.edu

## Abstract

### Background

Pathological mutations in Alpha-1 Antitrypsin (AAT) protein cause retention of toxic polymers in the hepatocyte endoplasmic reticulum. The risk for cirrhosis in AAT deficiency is likely directly related to retention of these polymers within the liver. Polymers are classically identified on liver biopsy as inclusion bodies by periodic acid schiff staining after diastase treatment and immunohistochemistry. However, characterization of the polymer burden within a biopsy sample is limited to a semi-quantitative scale as described by a pathologist. Better methods to quantify polymer are needed to advance our understanding of pathogenesis of disease. Therefore, we developed a method to quantify polymer aggregation from standard histologic specimens. In addition, we sought to understand the relationship of polymer burden and other histologic findings to the presence of liver fibrosis.

### Methods

Liver samples from a well-categorized AATD cohort were used to develop histo-morphometric tools to measure protein aggregation.

### Results

Whole-slide morphometry reliably quantifies aggregates in AATD individuals. Despite very low levels of inclusions present (0–0.41%), accumulation of globules is not linear and is associated with higher fibrosis stages. Immunohistochemistry demonstrates that fibrosis is associated with polymer accumulation and not total AAT. A proportion of patients were found to be "heavy accumulators" with a polymer burden above the upper 25% of normal distribution. Males had significantly more liver inclusions and polymer than females. These measurements also highlight interrelated phenotypes of hepatocellular degeneration and autophagy in AATD liver disease.

**Data Availability Statement:** Data analyzed in the publication are available in included spreadsheet. Specific algorithms used to analyze whole slide

images are included in the Supporting Information files.

**Funding:** Alpha 1 Foundation Professoship (MB), NIH NCATS University of Florida CTSA award L1TR001427 (VC, MB). The funders had no role in study design, data collection and analysis, decision to publish, or reparation of the manuscript. Dr. Collinsworth is currently affiliated with Advanced Pathology Solutions however she was faculty at University of Florida during the work included in this project. Advanced Pathology Solutions was not involved in the funding, design or execution of the work presented and did not provide relevant salary or other support to Dr. Collinsworth or any members of the research team during her work on this project.

**Competing interests:** Dr. Amy Collinsworth is currently employed at Advanced Pathology Solutions. However, her work on this project was completed while faculty at the University of Florida. Advanced Pathology Solutions was not a part of the design or funding of this project and she has not been involved in work on this project since her transition to her new commercial affiliation. Dr. Collinsworth's current affiliation does not alter our adherence to PLOS ONE policies on sharing data and materials and Advanced Pathology Solutions has had no research interest in our work.

## Conclusion

Quantitative inclusion analysis measures AAT accumulation in liver biopsy specimens. Quantification of polymer may identify individuals at risk for progressive disease and candidates for therapeutic interventions. Furthermore, these methods may be useful for evaluating efficacy of drugs targeting accumulation of AAT.

## Introduction

Alpha-1 antitrypsin deficiency (AATD) associated liver disease results from aggregates of polymerized alpha-1 antitrypsin. The most common disease variant is a single nucleotide missense mutation of glu-342-lys (PiZ) that causes improper folding and polymerization of AAT in the hepatocytes and unimpeded destruction of lung tissue during times of increased protease and inflammatory activity in the lung. The prevalence of homozygous individuals is approximately 1:3000 in the United States and over three million individuals are affected worldwide [1, 2]. Liver disease in AATD was originally described in 1969 as a phenomenon in infants who developed fulminant cirrhosis and abundant accumulation of intracellular inclusions of AAT glycoprotein that are Periodic Acid Schiff positive and diastase resistant (PAS-D) [3]. In the years since, it has become clear that adults also develop severe liver fibrosis caused by this "toxic gain of function" [4–6]. Retrospective analysis of liver transplant data has shown that decompensated liver disease requiring transplant is more frequent in adults than in children [7].

We recently showed that in a cross-section of non-cirrhotic adults with AATD, more than a third had clinically significant liver fibrosis, and fibrotic disease was strongly associated with PAS-D accumulation even when corrected for metabolic syndrome [8]. To establish this relationship, we developed a semi-quantitative grading score of PAS-D based on number of hepatocytes affected across a biopsy sample (none:0, rare: <5 cells, few: 5–20 cells, numerous: >20 cells). The goal of the scoring system was to establish a reproducible method for visual description of the polymer burden. In fact, the scoring system based on increased accumulation of AAT measured by PAS-D proved to be strongly associated with liver fibrosis. However, it likely underestimates the true polymer burden present within hepatocytes because not all polymerized AAT is captured by PAS-D staining. The contribution of the polymerized AAT that is outside of PAS-D aggregates to fibrosis is unknown. For these reasons and several other observations we made regarding PAS-D accumulation, more discriminative measurements of polymer burden are important. First, highly affected individuals appear to accumulate PAS-D globules in orders of magnitude greater than those with lower scores. Second, PAS-D aggregates were heterogenous in size and number, and the total area of the biopsy affected was variable. In a single biopsy sample, both small and large aggregates were present, and the distribution of globules was patchy, suggesting that hepatocytes accumulate AAT polymer differentially. Third, features of hepatocyte damage, inflammation, and steatosis were also associated with fibrosis, but their relationship with PAS-D accumulation requires further characterization.

To address these issues, we used baseline biopsy specimens from our previously described prospective clinical cohort of individuals with AATD to develop methods of quantitative analysis for PAS-D inclusions and to examine polymer specific immunochemistry of aggregated protein. Previous studies in the animal model of AATD have employed positive pixel counting and representative fields to quantify relative PAS-D accumulation in mice [9]. However, in

humans inclusions are often confined to few cells that are dispersed throughout a specimen. Liver biopsy tissue from early disease is very limited in quantity, so quantification by molecular methods cannot be reliably achieved while also preserving enough tissue for histological analysis. We created a series of Macro based tools for measuring the area, size, and number of PAS-D inclusions in the entirety of a digitally scanned specimen. We also created methods for measuring immunohistochemistry of AAT protein. Using these tools, we show that AAT polymer accumulation in hepatocytes is associated with liver fibrosis and that quantitative inclusion analysis identifies individuals with cellular changes associated with accumulation of aggregated proteins. We postulate these individuals are at risk for progressive liver disease.

## Materials and methods

### Patient recruitment

Patients were recruited and enrolled in the study between November of 2013 and October 2016 by three mechanisms: AATD patient registries, AATD patient outreach events, and clinical practice at UF. Eligible and unique individuals in two different patient registries (Alpha-1 Foundation DNA and Tissue Bank at the University of Florida and the Alpha-1 Foundation Research Registry, at the Medical University of South Carolina), were sent a letter describing the study and invited to participate. These national research registries are a confidential databases established with the purpose to promote research initiatives in AATD. It is estimated that over 2,500 (PI*ZZ or PI*SZ) individuals participate in the national registry. In this respect the patients are expected to be representative of the known North American adult population with PiZZ AATD. Inclusion criteria included ZZ individuals ages 21–71, from the US or Canada, who were willing to consent to a liver biopsy. Decompensated liver disease (ascites, variceal bleeding, hepatic encephalopathy, hepatocellular carcinoma) or previous liver or lung transplant were exclusion criteria.

**Human subjects.** In this study, we examined percutaneous liver biopsy samples from individuals with PiZZ AATD who are enrolled in a single center prospective cohort study [8]. Liver pathologists examined histopathology at the University of Florida. PAS-D accumulation was scored as P0 (no cells) with inclusions P1(<5 cells/biopsy) P2 (5–20 Cells/biopsy) or P3 (>20 cells). The University of Florida IRB approved all human subjects research in this study. The protocol was listed at clinicaltrials.gov (identifier NCT01810458) before recruitment. Additionally, PAS-D slides were examined from five explanted cirrhotic liver specimens from the Alpha-1 DNA and Tissue Bank. These specimens were fully de-identified and only specimen number and tissue diagnosis were known. Patients who participated in the DNA and tissue bank gave written informed consent for the use of explanted tissues. Study data were collected and managed using REDCap electronic data capture tools hosted at the University of Florida [10].

**Tissue preparation and staining.** We obtained biopsy samples using a 16 gauge BioPince full-core biopsy needle, the samples were immediately fixed in 10% formalin and transferred to UF Health Clinical Pathology Laboratories for paraffin embedding, H&E, PAS-D, PAS, and AAT immunohistochemistry (IHC) staining. The UF Molecular Pathology Core performed all other IHC. Supporting information contains additional details.

**Whole slide imaging.** Whole slide images were obtained using an Aperio CS2 slide scanner 20x/0.75 NA Plan Apo objective. 200x fields were scanned and processed into SVS formatted whole slide pyramidal tiff images. SVS formatted images were read and analyzed directly using ImageJ (FIJI build) [11–13].

**Quantification of PAS-D inclusions.** We quantified PAS-D stained 4 um liver sections using whole-slide images using tools in Image J (FIJI) distribution with its built-in image

processing plugins [14–16]. Specimens smaller than 12mm or with fewer than 10 portal tracts were not included in the analysis. Results represent data from n = 89 individuals. The first steps of the analysis select and measure the biopsy elements from background so that our measurements are normalized to the actual area of the tissue present on a slide. Then the specimens were background subtracted and subjected to thresholding, color deconvolution, binary pixel separation, and particle analysis. To compare computer imaging measurement to human observers, we used a random number generator to select 10 biopsies for manual measurements by two observers. Each observer measured individual globules by circular region of interest selection in the whole slide image at 200x. Proportionate area measurements were then compared with computerized algorithms with optimized cut-offs for size and circularity. See Supporting information for the full macro and more detailed methodology.

## Quantification of polymer and total AAT area

We used whole slide images of biopsies from the same cohort, stained with either total or polymer specific IHC against AAT. We measured Polymer specific AAT with a novel polymer specific monoclonal antibody which was previously created and described by Miranda et al. [17]. Similarly to the PAS-D images measurements, we normalized to the area of each biopsy. The Hematoxylin and Diaminobenzidine chromogen were separated using color deconvolution, and then positive pixel counts of Diaminobenzidine were measured as a proportion of total biopsy area. Importantly, these are estimates of histological area encumbered by polymer and not absolute quantity because chromogenic substrates and density of stain are not strictly stoichiometric with respect to protein quantity [18]. Further details including antibodies and dilutions are in the Supporting information.

**Analysis of microtubule associated protein light chain 3 beta (LC3B) immunohistochemistry and Annexin V Foci.** 20 subjects (10M, 10F) were selected for IHC analysis of LC3B and Annexin V. 10 subjects had evidence of hepatocellular degeneration as scored by a liver pathologist and 10 did not. Annexin V foci were counted in whole slide images and normalized to the area of the biopsy in mm$^2$.

**Statistical analysis.** Statistical analyses were performed using Graphpad Prism version 6.1. Gaussian data is presented as mean and SD and bivariate comparisons were performed using 2-sample t-tests. Non-Gaussian data as determined by D'Agostino- Pearson test is presented as median and interquartile range. Logarithmically distributed data for PAS-D globules were $y = \log_{10}(y)$ transformed and analyzed with one-way ANOVA with Tukey's Multiple comparison's test. Mann-Whitney U testing was used to make two group comparisons. Lins Concordance Correlation Coefficient was computed to determine the accuracy of computer algorithms compared to average hand measured globules by human observers.

## Results

### Clinical cohort

The cohort consisted of 94 subjects homozygous for the PiZ allele, 33 (35%) were male, and the median age of the subjects was 57. Histopathological characteristics and clinical correlates related to fibrosis for this cohort have been previously reported (8). 33 (35%) subjects had a METAVIR stage of F2 or greater. 89 (94.6%) individuals had PAS-D inclusions by pathological scoring and 93 (98.9%) had positive IHC for total AAT staining. Selected baseline characteristics are reported in **Table 1**. PAS-D slides from explanted cirrhotic PiZZ specimens were also examined for the purpose of measuring globules in the upper extent of disease, but no further clinical data were available from these specimens for comparison.

**Table 1. Selected baseline characteristics of the study population.**

|  | Baseline n = 94 |
|---|---|
| Age—median(IQR) | 57(52–63) |
| Sex M(%), F(%) | 33(35), 61(65) |
| METAVIR Score (0–3) | 26 / 35 / 27 / 6 |
| Fibroscan Median (kPa) | 6.4±2.8 |
| ALT (U/L) | 26±19 |
| AST (U/L) | 27±13 |
| GGT (U/L) | 40±78 |
| PT (s) | 14±0.78 |
| INR | 1±0.084 |
| PTT (s) | 31±2.7 |
| Platelets (1000/mm$^3$) | 224±59 |
| FVC% predicted | 88±19 |
| FEV1% predicted | 63±28 |
| FEV1/FVC | 70±21 |

ALT: Alanine aminotransferase, AST: Aspartate aminotransferase, GGT: Gamma glutamyl transferase, PT: Prothrombin time, INR: International Normalized Ratio, PTT: Partial thromboplastin time, FVC%: forced vital capacity % of predicted, FEV1%: forced expiratory volume % of predicted.

## PAS-D area is associated with liver damage

Quantitative inclusion analysis of biopsy specimens distinguishes globules from connective tissue and other PAS staining in biopsy slides (**Fig 1A, S1 Fig**). We tuned algorithms to determine which most accurately measures the % biopsy area occupied by inclusions. We found that size and number of globules per square millimeter was also correlated with liver disease. The median globule size for individuals was 10.31 um$^2$ (IQR 8.6–12.3) and the average size ranged from 6.2 to 41.2 um$^2$ though some individual globules were >100 um$^2$. Median globule number was 8.8 per mm$^2$ (IQR 2.2–21.1) with a range from zero globules to as many as 201 per mm$^2$. The % of total tissue occupied by globules provided the clearest way to account for differences in biopsy size and was most directly related to liver damage in the specimens.

The computer method was accurate in estimating globule area. We tested this by hand, measuring the biopsy and individual globules across 10 entire biopsy specimens using ROI measurement tools. We then tuned the object size and circularity parameters of our macro to develop a method which approximated human operators. We arrived at 5 algorithms that approximated the median area of accumulation. We tested these with Lins correlation coefficient and found that the coefficient between algorithm 4 and manual measurement was 0.93 (95% CI 0.75–0.99, p<0.0001). We then used this to examine all of the specimens (**Fig 1B**).

Inclusions represent a small percentage of the biopsy area but have significant associations with liver pathology. PAS-D measurements ranged from 0% (n = 1) to 0.41% of the biopsy area, the median % accumulation was 0.012% but varied widely with an IQR = 0.002–0.032%. Pathologist scoring of globules organizes accumulation into P0 (None), P1(<5 cells/biopsy), P2 (5–20 cells), or P3 (>20 cells). Our quantitative analysis agrees with increases in globules across pathological scores, but shows that the relationship between score and globule area is not strictly linear (**Fig 1C**). As predicted, accumulation increased with METAVIR stage, the median and interquartile range of PAS-D % area measurements were F0: 0.004% (0.0012–0.069), F1: 0.011% (0.003–0.2), F2: 0.019% (0.001–0.033), and F3: 0.056% (0.015–0.11). In explanted cirrhotic tissue F4, the median PAS-D % area measurement was 0.28% (0.21–0.73).

A

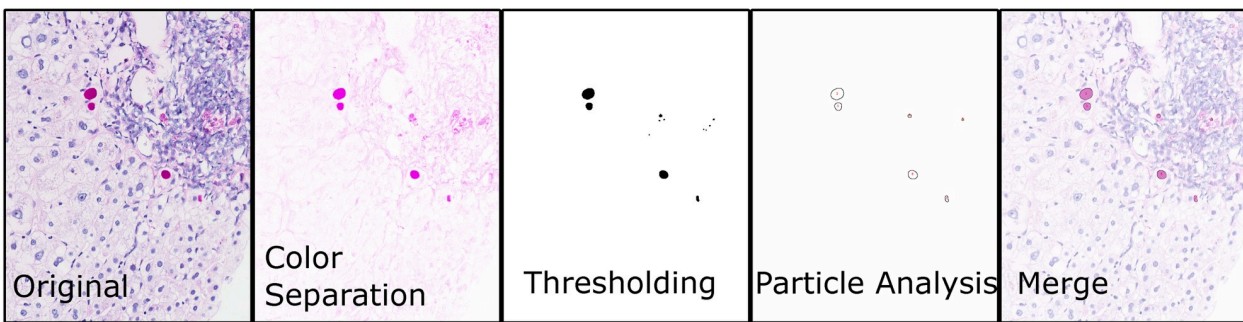

B

C

D

E

**Fig 1. Whole slide inclusion analysis defines the exponential nature of AAT accumulation as globules and its relationship with fibrosis. (A).** Stepwise representation of quantitative analysis of PAS-D inclusions 200 um scale bar **(B)** Selection of algorithm, points represent % area of a single biopsy occupied by inclusions measured by computer (C) or human observers (1, 2) bars represent median and IQR points represent % area measured from whole slide images. **(C)** Comparison between pathologist-based qualitative staging and quantitative analysis as median PAS-D % area and IQR **(D)** PAS-D % area compared to fibrosis stages bars represent median and IQR statistical testing via Kruskal-Wallis test with Dunn's multiple comparisons (* p<0.05, **P<0.01, ***p<0.001, ****p<0.0001) points represent % area measurements from individual subjects via whole slide images. F4 individuals are explanted cirrhotic liver from the Alpha 1 DNA and tissue bank Y- scale is Log10 **(E)** Separation of population based on quartiles and outliers (outliers defined as inclusion % area greater than or equal to Q3+1.5*IQR).

There was a 2.75-fold change between no fibrosis and F1, a 1.72-fold change between F1 and F2, and a 2.95-fold change between F2 and F3 as shown in (**Fig 1D**). The data is presented on a log scale to emphasize the variability in accumulation between individuals at each fibrosis stage. The data was tested for normality using D'Agostino-Pearson testing, and it was not normally distributed. Kruskal-Wallis Testing showed significant difference between median number of globules across fibrosis stages Kruskal-Wallis statistic was 27.64 and the p-value was <0.0001 post-Hoc analysis with Dunn's multiple comparisons test showed significant increases in PAS-D % area between F0vF2, F0vF3, and F1vF3. Comparisons between the biopsy tissue and explanted cirrhotic tissue were significant for F0vF4, F1vF4, and F2vF4, but not F3vF4. When evaluating PAS-D % area without regard to fibrosis score, a group of PiZZ individuals were identified as "high accumulators" and are in the upper quartile or outliers with accumulation that is >1.5x the interquartile range of the population (**Fig 1E**). We hypothesize that these individuals with the greatest accumulation are at risk for progressive liver fibrosis, which can only be shown over time.

## Polymer accumulation and liver fibrosis in AATD

Although PAS-D inclusions are classic histological findings in AATD, AAT also accumulates in the liver as polymers that are not part of PAS-D inclusion bodies. The polymeric AAT can be detected using monoclonal antibodies raised against AAT polymer as described previously (17). The other stainable fraction of AAT is the total fraction which includes pas-d globules polymers and non-polymeric AAT, we refer to this as total AAT. The relationship between polymer and globules has not been previously examined in non-cirrhotic individuals with AATD. We hypothesized that quantitative histological analysis of these polymers could also identify individuals with increased risk of liver fibrosis. We evaluated IHC slides from the same biopsies using a positive pixel counting algorithm to calculate the % area positive for total and polymer specific immunohistochemistry (**Fig 2A and 2B**). We examined polymer and total AAT in the same subjects as PAS-D. Polymer IHC could not be performed on 4 subjects in the original cohort and was not possible in explanted cirrhotic tissue due to technical issues with specimens collected and preserved with different methods and years apart. The median area occupied by total AAT staining is 19% but varied between individuals (0.41–56%). Reliable staining of total AAT was difficult because specimens had large amounts of AAT total staining and positive edge artifacts were frequent. Seven specimens were not included in the analysis as they lacked size or had significant staining artifacts in recuts. Polymer % Area showed a median staining of 1.8% that was variable (IQR 0.067%-12%). Polymer area was increased with higher fibrosis stages. The groups were not normally distributed as determined by D'Agostino-Pearson testing. Kruskal-Wallis testing with Dunn's multiple comparisons showed significant differences in polymer accumulation between fibrosis stages p<0.0002 and Kruskal Wallis statistic was 19.29, post-hoc analysis was then performed F0 had significant differences with F2 and F3, F1 was different than F3. However, Total IHC measurements had no statistically significant relationship with fibrosis (**Fig 2C and 2D**). Polymer area also identified 5 of the 8 individuals who were outliers in PAS-D analysis. Although three

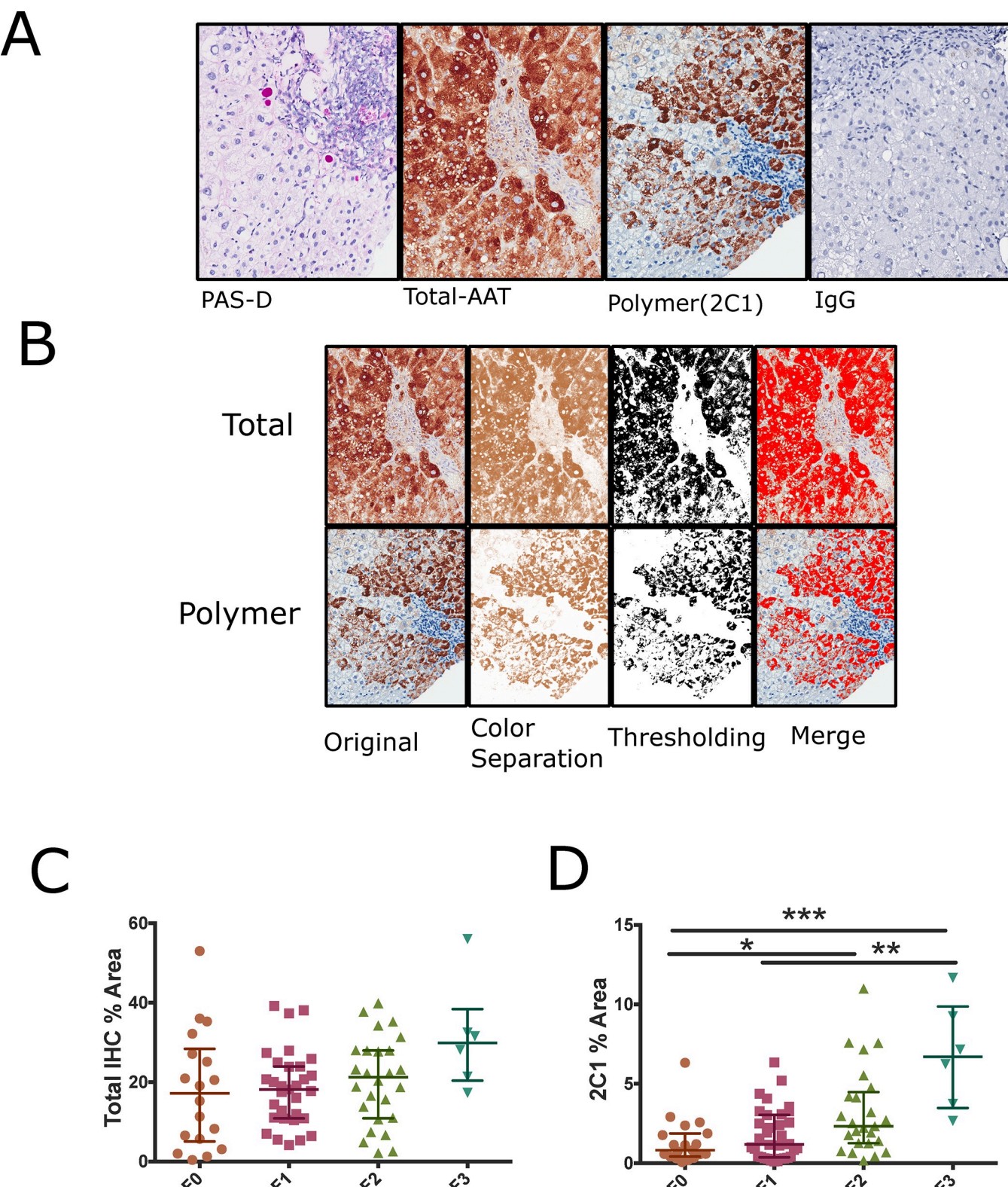

**Fig 2. Polymer IHC % area is a complementary measure of AAT accumulation.** (A) Semi-serial sections of PAS-D, Total and Polymer AATD with IgG control (B) Representation of IHC threshold-based quantification of AATD liver (C) (D) Quantification of total and polymer specific immunohistochemistry across METAVIR fibrosis stages Kruskal-Wallis testing with Dunn's multiple comparisons *p<0.05, **p<0.01. ***p<0.0001 bars represent median and IQR, points represent measurement from individual whole slide images.

individuals did not have discernable PAS-D globules, all subjects had Polymer detected by immunohistochemistry.

## AAT accumulation is associated with male sex

In animal models of AATD, male sex is associated with increased AAT aggregation (17). Sex differences in AAT aggregation in humans has not been previously evaluated, so we sought to address this question. 32 males and 57 females had biopsies sufficient for analysis. AAT accumulation as measured by both median PAS-D% area and Polymer % area was higher in males. Median PAS-D % area in males was 0.0195% vs 0.0100% in females (p = 0.0167, **Fig 3A**). Polymer area was also higher in males than females (2.399% vs 1.96%, p = 0.0456) **Fig 3B**. Furthermore, the outlier group of "high accumulators" was evaluated by sex, and nearly all of them

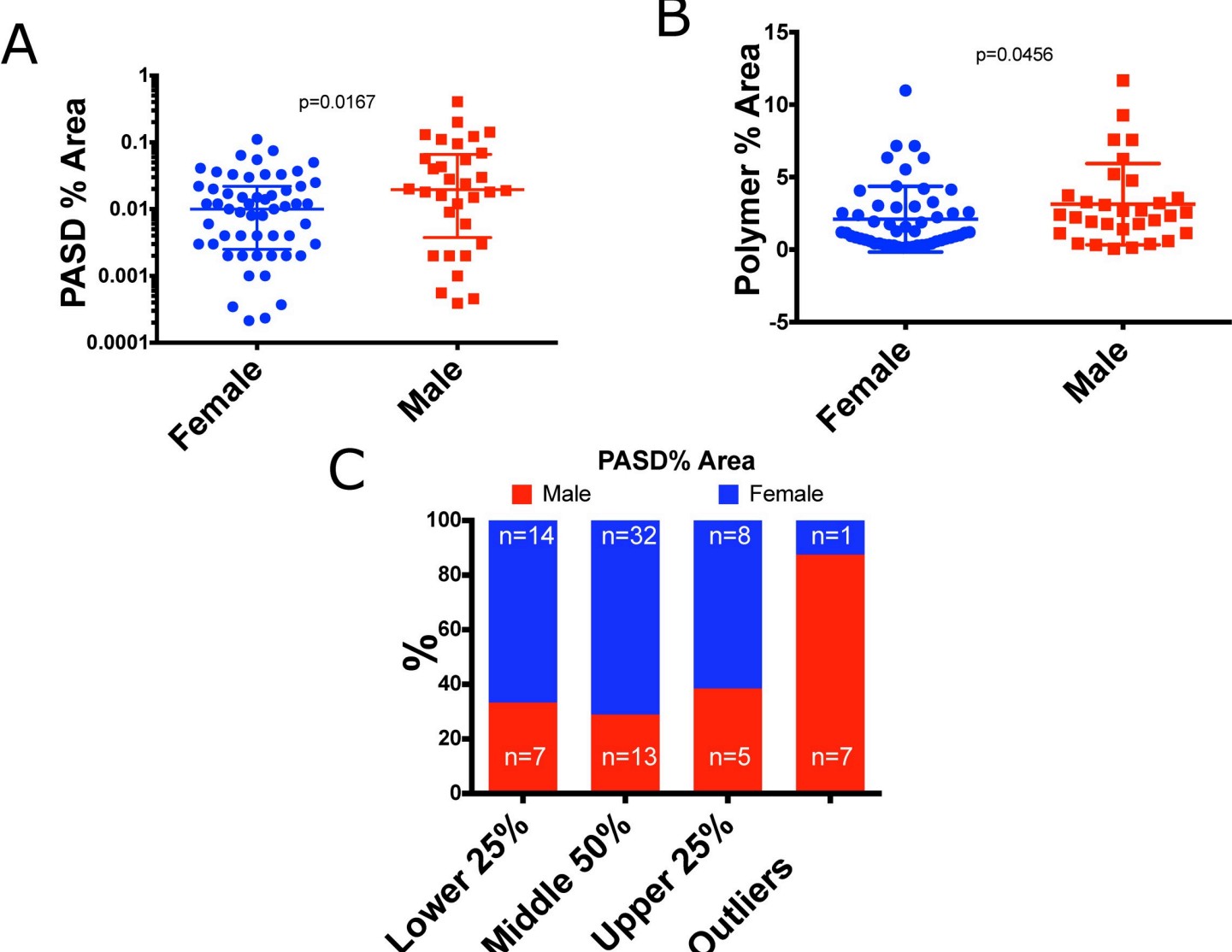

**Fig 3. Males are more affected by AAT accumulation.** (A) Quantitative inclusion analysis in male and female subjects p-value calculated via Mann-Whitney U-test (B) Polymer % area in male and female subjects p-value calculated via Mann-Whitney U-test (C) Distribution of PAS-D inclusions between males and female distribution calculated from PAS-D % area and outliers are individuals with PAS-D % area greater than or equal to Quartile 3 + 1.5*IQR.

were male (87.5% n = 7) **Fig 3C.** This observation is interesting because male sex is reported as a clinical risk factor for cirrhosis (18). It can be hypothesized that higher accumulation of AAT in males provides some biological plausibility for the increased fibrosis risk.

## Hepatocellular degeneration in aggregate related liver disease

The presence of inflammation and hepatocytes with mild hydropic changes, abundant glyco-gen, and scattered macrophage activation termed "hepatocellular degeneration" are histologi-cal findings intertwined with fibrosis and AAT accumulation. (**Fig 4A**) shows how these characteristics affect groups of hepatocytes. When hepatocellular degeneration was present in the biopsy sample, the median fibrosis scores were increased and PAS-D accumulation was 7.5 fold higher (PAS-D% Area = 0.030 vs. 0.0040, p<0.0001) as shown in (**Fig 4B and 4C**). Poly-mer % area was 2 times greater (2.52 vs. 1.13, p = 0.0001) when hepatocellular degeneration was present (**Fig 4D**). Polymer was present in hepatocytes with early cytoplasmic clearing and signs of degeneration. However, some groups of hepatocytes had cleared cytoplasm and nega-tive polymer staining (**Fig 4E**).

We hypothesize that the observed hepatocyte clearing is connected to autophagy of cellular constituents. AATD has been previously associated with autophagic responses in humans, ani-mal models, and cell culture systems [19]. In the PiZ animal model of AATD, boosting autop-hagy clears AAT inclusions [20, 21]. We used LC3B immunohistochemistry to highlight macro-autophagy in biopsy specimens from individuals with high (n = 10) and low (n = 10) AAT aggregation [22]. LC3 IHC was positive in the cytoplasm, nuclei, and in discrete puncta. AAT inclusion bodies displayed circumferential LC3B staining. Individuals with low accumu-lation of AAT had faint LC3B staining and a paucity of LC3B puncta. Hepatocytes with cleared cytoplasm and moderate to heavy polymer accumulation displayed abundant LC3 puncta (**Fig 5A**). Furthermore, we hypothesized that presence of degeneration could be associated with early apoptotic cell death. Annexin V staining was used as a surrogate marker for early apopto-sis. Positive foci were more abundant in individuals with hepatocellular degeneration (0.90 ±0.32 vs 0.58±0.28 foci/mm$^2$, p = 0.0366). However, these foci were not localized directly to hepatocytes with cytoplasmic clearing (**Fig 5B**).

## Discussion and conclusions

AATD liver disease is an occult phenomenon with the potential for progression to liver cirrho-sis. The driving force for liver injury is the retention and accumulation of AAT within hepato-cytes, which can be semi-quantitatively measured by PAS-D staining of liver tissue. More precise quantification of polymerized AAT is possible with molecular methods, but these require tissue destruction and are not standardized because obtaining adequate liver tissue in a rare disease population such as AATD is difficult. In this work, we describe a describe a method for quantification of AATD globules in liver biopsy specimens using an imaging tech-nique that can be applied easily applied to a clinically obtained specimen. The technique described here allows for precise characterization of polymer accumulation and the relation-ship with fibrosis. Specifically we showed there are statistically significant differences in PAS-D globules and Polymer IHC between people with lower fibrosis score and people with higher fibrosis score. There were not differences in total AAT Immunohistochemical staining through different fibrosis stages.

Our characterization of early liver disease in AATD showed that there is significant hetero-geneity in accumulation of AATD globules and polymer despite a homogenous mutational background. The quantitative analysis highlights that globules occupy only a very small por-tion (<0.5%) of a given liver sample despite all individuals having PiZZ AATD. As expected,

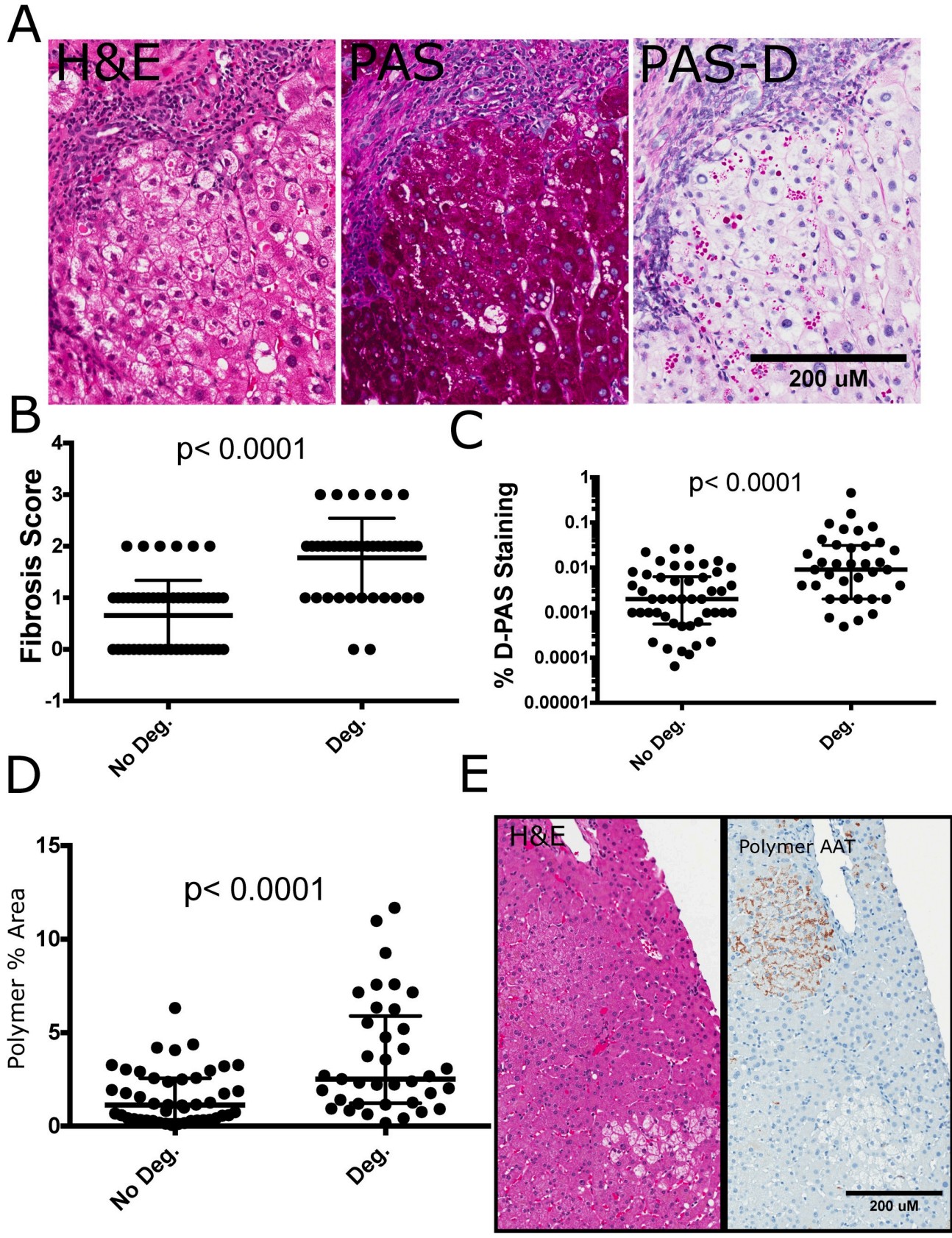

**Fig 4. Hepatocyte degeneration is a predominant feature of AATD related polymer accumulation.** (**A**) Hepatocyte degeneration is a predominant feature of AATD liver disease. H&E slides (Left panel) show partial cytoplasmic clearing with or without the presence of inclusion bodies, these cells are glycogenated as evidenced by PAS stain (middle panel) and some but not all degenerate hepatocytes have PAS-D AAT inclusions (right panel) (**B**) (**C**) (**D**) Degeneration is associated with higher median fibrosis scores Mann-Whitney U Test, p<0.05 considered significant. bars represent median and IQR n = 94 fibrosis, n = 89 PAS-D, Polymer % Area (**E**) Polymer staining was present in partially cleared hepatocytes but absent from cells with more severe degeneration. Scale bars = 200um.

we showed that PAS-D aggregates are related to liver fibrosis. Importantly, the more precise method for quantification with PAS-D% area demonstrated that the relationship between fibrosis stages and PAS-D globules is logarithmic rather than strictly linear. Furthermore, there are significant differences between accumulation of polymer measured by PAS-D % area between the lowest and highest stages of fibrosis. However, high accumulation of polymer was also seen at low fibrosis stages, so it correlates poorly in early fibrosis. We hypothesize that these individuals with high accumulation and low fibrosis are early in the natural history of disease. An alternate explanation is other unknown factors prevent progression of fibrosis despite accumulation of AAT. This is a limitation of this cross-sectional sample and can only be addressed with long term follow up.

The accumulation of total-AAT staining in excess of polymer and PAS-D helps to show that AAT, in multiple forms, is retained in the liver. As anticipated, the % area of biopsy with polymer when measured by IHC (median 1.8%) was greater than when measured by PAS-D (median 0.012%) because the antibody stains AAT polymer within inclusions as well as the polymer outside of inclusions. Regardless, polymer and inclusions are both associated with fibrosis. If changes in polymer % area are sensitive to detect decreases in AAT production after therapeutic intervention remains to be determined. It is important to note that these data are based on histological data. Ideally enough tissue would be available for quantitative analysis though direct and destructive methods such as mass spectrometry. However, additional specimens for this purpose were not obtained due to safety and technical considerations in our cohort. Instead, our data are based on paraffin imbedded specimen commonly available with clinical biopsies. Thus, quantitative analysis could reasonably be performed on both recent or archived specimens from patients who have had liver biopsies to determine the extent of AAT aggregation.

In addition to aggregated protein, the phenotype of hepatocellular degeneration is closely related with fibrosis. Individuals with degeneration have more inclusions, polymer, and higher fibrosis stages. However, these damaged cells appear to have less polymer. Similar age-related phenotypes in the PiZ mouse model suggest that degenerate cells may be pre-malignant [23]. There is evidence of autophagic responses in humans, mice and in-vitro models of AATD. These studies show lysosomal degradation pathways are active, but endogenous autophagic activity is insufficient to prevent polymer aggregation [19, 24]. Indeed, the PiZ animal model has an autophagic response associated with fibrosis, but enhancing autophagy decreases accumulation of AAT and may prevent fibrosis [9, 20, 21, 25]. However, the functional consequences of the autophagic response in cells after they clear polymer have not been extensively studied. In this population, we observed signs of an active autophagic response that was also associated with hepatocyte damage. These data raise the question of whether or not long-term autophagic responses, or exogenously stimulated autophagy as a therapy will halt or drive liver disease related hepatocyte degeneration in afflicted individuals.

In this cohort, male sex was not an independent predictor of fibrosis after adjusting for presence of metabolic disease [8], although other studies have suggested males have increased risk for cirrhosis. In prior studies transgenic male animals have increased AAT accumulation that can be partially replicated in female mice supplemented with androgens [26]. Our data

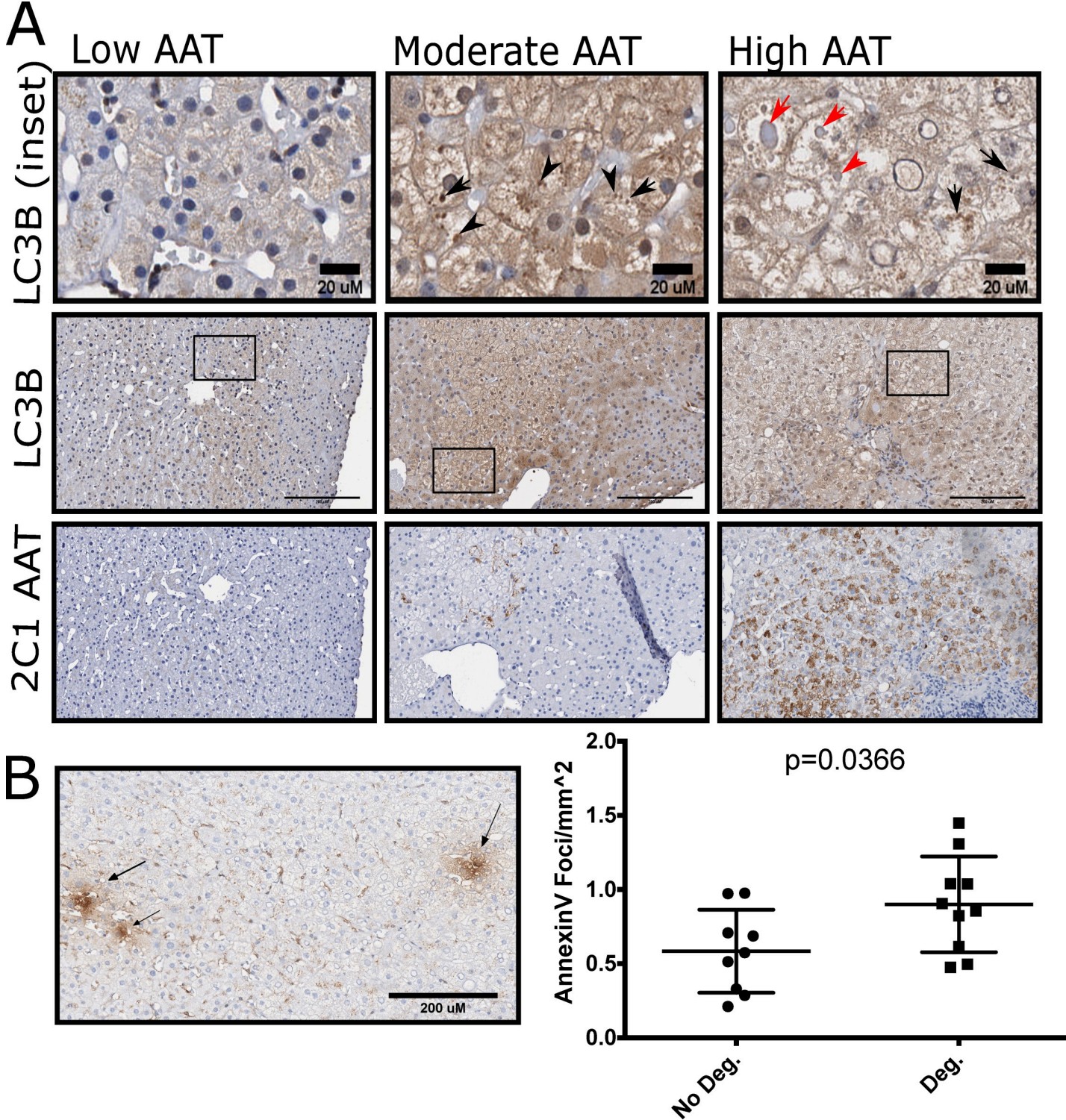

**Fig 5. Autophagy connects toxic gain of function to cellular degeneration. (A)** Representative images from n = 20 PiZZ subjects comparing polymer accumulation, LC3B staining and hepatocyte degeneration. The top row is an inset of LC3B staining from the middle row (marked with a black box). AAT polymer staining in the same region is presented in the bottom row. individuals with moderate and high accumulation of AAT. Black Arrows represent puncta of LC3B in degenerate hepatocytes. Red Arrows represent staining of LC3B around AAT inclusions. **(B)** Representation and quantification of Annexin V Foci (Black Arrows) were more prominent in individuals with degeneration, Bars represent mean and standard deviation p-value is calculated from student's t-test.

show that men and women both have significant accumulation of both polymer and inclusions, but overall men had more AAT aggregation as measured by PAS-D % area. Although there is a predilection for men to have more aggregates, AAT accumulation is variable in both sexes. Many women with AATD have both fibrosis and heavy accumulation. The biological causes of these differences have not been elucidated and need further investigation especially with regards to differences in response to targeted therapeutics (S1 Graphical abstract).

Advances in imaging and computer technology in the last decade allow us to glean quantitative insights from tissue specimens using pixel data encoded by the in-situ dynamics of an individual's tissue [27]. In-vitro work shows that luminal motility in the endoplasmic reticulum is compromised in AATD [28]. There are certainly technical issues which may exist in the application of these methods to other sample groups. Specifically, the samples in our study represent a relatively controlled group of tissues in that they were all collected and processed in the same fashion. Artifacts and sample processing variances may complicate the direct application of these methods or require retuning of parameters to accommodate differences in technique.

Current therapeutic trials focus on inhibiting accumulation of AAT in liver by inducing autophagy or silencing expression of AAT before a patient develops decompensated cirrhosis. The methods we provide in this work are intended to advance the quantitative value of histological samples for research. These tools may also be useful for evaluating the efficacy of therapeutics aimed at reducing accumulation of AAT. We used these algorithms and a systematic analysis of histological characteristics to confirm previous suppositions about the dynamics of AAT and provide quantitative insight into how toxic gain of function may predict disease progression. However, it remains to be seen if high accumulators have longitudinal progression of disease. We may have additional insights into this important question as the study subject's complete longitudinal follow-up.

## Supporting information

**S1 Text. Supporting methods.**
(DOCX)

**S1 Fig. Representative steps and processing of inclusion analysis. (A)** Positive pixel counting in PAS-D slides over-represents accumulation in humans **(B)** Representation of whole slide processing low resolution **(C)** Representation of steps involved in inclusion analysis.
(TIF)

**S1 Data. Path data.**
(XLSX)

**S1 Graphical abstract.** Top panel, whole slide imaging of PAS-D and polymer specific IHC, Middle panel, median PAS-D, Polymer and Total IHC shows the different fractions and quantitative analysis identifies specific individuals who has high and very high AAT accumulation. Bottom panel, High accumulators identified by inclusion analysis may be at higher risk of progressive disease and theoretically may benefit from future interventions.
(TIF)

## Acknowledgments

The UF Molecular Pathology Core provided histological services and consultation on methods development and slide scanning.

We are grateful to William Dunn, PhD his help on autophagy and histological analysis of hepatocellular degeneration.

## Author Contributions

**Conceptualization:** George Marek, Chen Liu, Mark Brantly, Virginia Clark.

**Data curation:** George Marek, Chen Liu, Mark Brantly, Virginia Clark.

**Formal analysis:** George Marek, Virginia Clark.

**Funding acquisition:** Mark Brantly, Virginia Clark.

**Investigation:** George Marek, Mark Brantly, Virginia Clark.

**Methodology:** George Marek, Amy Collinsworth, Chen Liu, Mark Brantly.

**Project administration:** Mark Brantly, Virginia Clark.

**Resources:** George Marek, Amy Collinsworth, Mark Brantly, Virginia Clark.

**Software:** George Marek.

**Supervision:** Chen Liu, Mark Brantly, Virginia Clark.

**Visualization:** George Marek, Amy Collinsworth, Chen Liu, Virginia Clark.

**Writing – original draft:** George Marek.

**Writing – review & editing:** George Marek, Mark Brantly, Virginia Clark.

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
