## [Decision Letter · Decision Letter 0]

18 Feb 2021

PONE-D-21-03416

Quantitative measurement of the histological features of Alpha-1 Antitrypsin Deficiency-associated liver disease in biopsy specimens

PLOS ONE

Dear Dr. Marek,

Thank you for submitting your manuscript to PLOS ONE. After careful consideration, we feel that it has merit but does not fully meet PLOS ONE’s publication criteria as it currently stands. Therefore, we invite you to submit a revised version of the manuscript that addresses the points raised during the review process.

While both reviewers agreed that this is a carefully performed study, several issues were raised that should be addressed prior to a publication.

We look forward to receiving your revised manuscript.

Kind regards,

Pavel Strnad

Academic Editor

PLOS ONE

Journal Requirements:

2) In your methods, please clarify if the biological samples used in your study were completely de-identified before researchers accessed the scirrhotic liver specimens from the Alpha-1 DNA and Tissue Bank.

3)  In your Methods section, please provide additional information about the participant recruitment method in the original cohort study and the demographic details of the participants. Please ensure you have provided sufficient details to replicate the analyses such as:

a) the original recruitment date range (month and year),

b) a description of any inclusion/exclusion criteria that were applied to participant recruitment,

c) the name of the institution patients were recruited from,

d) a statement as to whether your sample can be considered representative of a larger population, and

e) a description of how participants were recruited.

4) Please upload a copy of Figure 6, to which you refer in your text on line 436. If the figure is no longer to be included as part of the submission please remove all reference to it within the text.

5) Please include captions for your Supporting Information files at the end of your manuscript, and update any in-text citations to match accordingly. Please see our Supporting Information guidelines for more information: http://journals.plos.org/plosone/s/supporting-information.

6) Thank you for stating the following in the Competing Interests section:

[The authors have declared that no competing interests exist.].   

We note that one or more of the authors are employed by a commercial company: Advanced Pathology Solutions

i. Please provide an amended Funding Statement declaring this commercial affiliation, as well as a statement regarding the Role of Funders in your study. If the funding organization did not play a role in the study design, data collection and analysis, decision to publish, or preparation of the manuscript and only provided financial support in the form of authors' salaries and/or research materials, please review your statements relating to the author contributions, and ensure you have specifically and accurately indicated the role(s) that these authors had in your study. You can update author roles in the Author Contributions section of the online submission form.

ii. Please also provide an updated Competing Interests Statement declaring this commercial affiliation along with any other relevant declarations relating to employment, consultancy, patents, products in development, or marketed products, etc. 

Reviewers' comments:

Reviewer's Responses to Questions

**Comments to the Author**

1. Is the manuscript technically sound, and do the data support the conclusions?

Reviewer #1: Partly

Reviewer #2: Partly

2. Has the statistical analysis been performed appropriately and rigorously? 

Reviewer #1: Yes

Reviewer #2: No

3. Have the authors made all data underlying the findings in their manuscript fully available?

Reviewer #1: Yes

Reviewer #2: Yes

4. Is the manuscript presented in an intelligible fashion and written in standard English?

Reviewer #1: Yes

Reviewer #2: Yes

5. Review Comments to the Author

Reviewer #1: In their study Marek et al. developed a method to quantify polymer aggregation in alpha-1 antitrypsin deficiency from standard histological specimens to improve our understanding of disease pathogenesis. In addition, they sought to understand the relationship between polymer burden and the presence of liver fibrosis, and evaluated gender differences.

The authors used 94 PiZZ individuals of their well-characterized AATD cohort to develop histo-morphometric tools for the quantitative measurement of PAS-D inclusions and examined polymer specific immunhistochemistry of aggregated protein. These tools demonstrated that the relationship between fibrosis stages and PAS-D globules appears to be logarithmic rather than linear and the area of biopsy occupied by PAS-D globules increases with fibrosis stage. IHC staining showed that liver fibrosis is associated with polymer accumulation and not total AAT.

Since there are only semi-quantitative grading scores and promising new therapeutic options are currently uprising, the results of this study are of great interest. Nevertheless, there are a few weaknesses, that diminish the value of the presented data:

1. As artifacts and processing deviations may confound and complicate this technique, the transferability to samples collected in a slightly different fashion seems to be complicated.

Moreover, the biological role and significance of histological techniques should not be overestimated. Nevertheless, the sample size is of course impressive given that AATD belongs to the rare diseases.

2. Are there already any longitudinal data (any decompensated liver disease etc.) available on the examined individuals to estimate the prognostic importance?

3. Next to male sex, there are other risk factors whose presence is already known to contribute to the development of liver disease, such as obesity, higher age, or the presence of diabetes mellitus. Are there also differences in AAT accumulation when comparing subgroups of the factors mentioned?

Minor comments:

- Page 19, line 374 ff. “The finding of PAS-D aggregates in an individual with AATD who has early liver fibrosis indicates a of toxic gain of function that is meaningful.” – remove “of”

- Page 21, line 436 “figure 6, graphical summary”- somehow I cannot find figure 6…..

- Figure 2A: “Polymer(2C1)” – reduce the rear bracket

Reviewer #2: The manuscript by Marek et.al. entitled “Quantitative measurement of the histological features of Alpha-1 Antitrypsin Deficiency-associated liver disease in biopsy specimens” is a descriptive paper of alpha-1 antitrypsin polymers designed to enhance our understanding of this previously sidelined aspect of AATD associated liver disease. The paper can be improved by addressing the following issues:

Major

1. I was impressed with the very low (0-0.41%) “levels” of inclusions. It seems like the more specific measurement of the denominator in mm2 or other specific measurement metric would make this a more universally accepted method.

2. The introduction on line 96-98 discusses the heterogeneity of PAS-D globules that suggested that hepatocytes accumulate AAT polymer differentially. This is not known and you should prove this in the current paper.

3. In the introduction lines 100-101 you note that “PAS-D aggregates represent only a fraction of the total AAT polymer present in hepatocytes”. I believe the more accurate statement is that PAS-D aggregates are made up of both non-polymeric and polymeric forms of retained AAT. Therefore, this statement needs correction.

4. Line 152. It is not at all clear what is meant by “measure the specimens”. Does this mean that analyzed areas were normalized for the slide area visualized? If so, were the scoring systems for similarly normalized when counting PAS-D inclusions?

5. I would assume that more than one slide per patient were used to minimize random effects? Was this done? How many slides, how many fields?

6. Why would you normalize Annexin V to mm2, but the other analyses are left to counts on a 200x field? (Line 180)

7. I am getting lost in the analysis numbers. Were all 94 samples subjected to globule analysis but only 20 were chosen for IHC of polymer?

8. “The variability of accumulation… was exponential” suggests that analysis should be done using a log transformation of the data for all the analyses. I do not see that this was done.

9. Concordance is usually measured using a Kappa statistic for categorical values or Lins concordance correlation (or others) for continuous data. Which was used? Put it in the methods. Confidence intervals would then denote whether there was difference between computer and human observers.

10. Just when I think I have a handle on your data, you throw in data from explanted cirrhotic F4 tissue in line 235. Where did this come from? Do you want to go back and put this in the methods and do some comparisons?

11. When I look at the graphs for PAS-D% between FO and F2, it looks like the data is normally distributed. I will need a test for normality to fail before I would allow a Kruskal-Wallis with post hoc Dunn’s to be used. The alternative explanation is that PAS-D% is poorly correlated with early stage measures of fibrosis until F3 and F4 are encountered. This is clearly an important point since in the discussion (line 387), this is one of your major conclusions. I disagree with this conclusion after looking at your data. The alternative is that a small subset of individuals have a very high PAS-D burden associated with fibrosis.

12. Line 387 suggests you have moved from association analyses to cause and effect thinking. You can speculate, but not conclude such.

13. Figure 1 graph E adds no value in my opinion. Please justify what this adds.

14. As I read your numbers, the attempt to quantify polymer was in 20 patients and in 7 patients this was not possible for technical reasons. Then the following is quoted “Polymer area was increased with higher fibrosis stages. However, Total IHC measurements had no statistically significant relationship with fibrosis”. Just say that you could not find a relationship between polymer numbers and fibrosis if you cannot prove it statistically.

Minor

1. Abstract line 54 remove “a”

2. Introduction: line 65 ”Alpha-1 Antitrypsin Deficiency” should not be capitalized. This is a disease state similar to diabetes, coronary artery disease, etc.

3. Introduction Line 66: Alpha-1 antitrypsin is a protein and should not be capitalized similar to elastin, collagen, interleukin-6, etc.

4. Line 78 …requiring transplant “is” more frequent in adults.

5. Line 85 and again on line 128. This scoring system requires that you state per high powered field and the magnification at which the slide was stained.

6. Sentence 103 is awkward and should be reworded.

7. Sentence 110 should be broken into 2-3 sentences.

8. 141: You should write out immunohistochemistry (IHC) before using the abbreviation

9. Line 177: You should write out LC3B the first time of use.

10. Line 195: Do you mean a positive IHC for AAT polymer?

11. Table 1. FEV1/FVC does not have %. The primary ratio should be used.

12. Sentence 397 is awkward and should be rewritten.

6. PLOS authors have the option to publish the peer review history of their article (what does this mean?). If published, this will include your full peer review and any attached files.

Reviewer #1: No

Reviewer #2: No

---

## [Author Response · Author response to Decision Letter 0]

30 Apr 2021

Dear Dr. Strnad and Review Committee,

Thank you for reviewing our work and offering us the opportunity to incorporate the comments and revisions requested by the review committee. Our paper entitled Quantitative measurement of the histological features of alpha-1 antitrypsin deficiency-associated liver disease in biopsy specimens uses whole slide imaging to study baseline histologic data from a large longitudinal cohort of people who have PiZZ alpha-1 antitrypsin deficiency. We demonstrate and provide new tools which can be used to estimate severity of alpha 1 antitrypsin accumulation in standard pathology specimens. We have addressed each reviewers major and minor revisions as below. We have also made editorial revisions as noted.

Editorial Revisions

1) We have addressed the style requirements and file naming for PLOS ONE

2) We have clarified this in human subjects section, see line 141

3) We have included an additional participant recruitment section which details this information, Table 1 provides basic demographic information for the population and more extensive details have been previously published in our original descriptive work [8]

4) Figure 6 (graphical summary) has been uploaded

5) We have updated the supplementary material as requested

6) With respect to Dr. Collinsworth’s affiliation with a commercial histopathology group:

Financial Interests Statement:

Dr. Collinsworth is currently affiliated with Advanced Pathology Solutions however she was faculty at University of Florida during the work included in this project. Advanced Pathology Solutions was not involved in the funding, design or execution of the work presented and did not provide relevant salary or other support to Dr. Collinsworth or any members of the research team during her work on this project.

Competing Interests Statement:

Dr. Amy Collinsworth is currently employed at Advanced Pathology Solutions. However, her work on this project was completed while faculty at the University of Florida. Advanced Pathology Solutions was not a part of the design or funding of this project and she has not been involved in work on this project since her transition to her new commercial affiliation. Dr. Collinsworth’s current affiliation does not alter our adherence to PLOS ONE policies on sharing data and materials and Advanced Pathology Solutions has had no research interest in our work.

Reviewer #1

Reviewer #1 had specific major concerns related to the generalizability of our methods to wider use and was interested in longitudinal and comparisons of PAS-D accumulation with other known metabolic diseases, sex and aging.

1. We certainly agree that histological analysis should not be overestimated. One advantage of this method is that it was designed around a standard stain (PAS-D) which is used across many institutions and is standard practice for evaluating liver biopsies of most people who have had a liver biopsy should have a PAS-D slide archived. Even the path specimens in our cohort were not strictly without differences in hue and stain intensity which is why background subtraction tools, color deconvolution and binary operations are used. 

2. We have some data available, including yearly non-invasive measurements which suggest association between worsening transaminases and liver stiffness measurements and AST to platelet ratio index which we plan to analyze together with other clinical markers together as part of forthcoming publications.

3. This is a very appropriate question, the answer is yes, but we have already published these relationships in our original paper discussing this same cohort and with our qualitative scores. 

Minor comments:

- We made these changes as requested.

Reviewer #2

Reviewer #2 had several major revisions and comments. One important feature of these comments which needs to be clarified is the nature of our measurements. There are several questions discussing the data in terms of standard methods using high-powered fields. It is very important to note that we are providing a method to estimate globule area in whole slide images. This is extensively discussed in our methods and supplementary material and is essential to interpretation of our data. We have addressed this and additional concerns including requests for additional statistical analysis and details as below.

1. The data is presented as % area because we analyzed whole slides and globules have both size and number that contribute to how much of a biopsy is occupied.

2. We have added additional data about the size and number of globules including their confidence intervals which helps to quantify these differences. Additionally, our histologic polymer analysis shows very clearly that only a portion of any given individual’s liver is occupied by polymer and differences in hepatocellular accumulation of AAT are discussed with relation to hepatocyte degeneration and autophagy in results and discussion of this paper. Additionally, our original publication on the clinical characteristics also describes the zonal distribution of globules and has already been published.

3. With respect to the relationship between globules and polymer, the original statement is correct and is supported by our data showing polymer staining occupies a larger proportion of the biopsies than PAS-D aggregates alone. PAS-D aggregates are large aggregates of polymer, but not all polymer is within a PAS-D aggregate. Total AAT antibodies stain both polymeric and non-polymeric AAT, and total AAT staining did not have a relationship with other aspects of liver pathology. See also Fig 6 which we added in our revised manuscript.

4. We added additional statements in the methods and supplement to help clarify this point. The method is based on a whole slide analysis, which means the whole slide is scanned initially, which includes background white space. Our analysis measures the area of the biopsy based on total pixels that are positive after background subtraction. The measurements of globules within this area are then normalized the total area of each biopsy so that the size of the biopsy ultimately does not impact the quantification by this method. 

5. This is whole slide imaging, so every field on a biopsy is counted. This is a seminal feature of our work in that the quantification of PAS-D is not biased by the number or location where fields are chosen. 

6. The counts are whole slide analysis so this data represents all the annexin foci in the entire biopsy and divided by the size of tissue in mm2 so it is not a representation of a number of fields of view.

7. Polymer IHC was examined on the same population as PAS-D. The individual point values are shown in figure 2. Four samples couldn’t be analyzed because of tissue constraints, and total staining has high background and artifacts, so 7 samples couldn’t be used for this analysis.

8. We did do these analyses as well. Log transformed data reduced the skew and created a normal distribution for running ANOVA but this did not change the overall conclusions of the data.

9. We developed the algorithms using lins statistic comparing the observers to five putative algorithms and chose the best from here. I have added this information to the methods and results.

10. The explanted specimens used are archived slides some of which come from early days of the DNA and tissue bank. Very few of these are available from explanted cirrhosis tissue and they patients are not well characterized. For our purposes we examined them to measure PAS-D to provide the upper extremes of disease because our cohort of well characterized biopsies excluded decompensated cirrhosis. We do not have further data to make comparisons with these specimens.

11. The data looks normally distributed because it is placed on a log scale to emphasize the nature of variability between individuals. The groups failed D’Aogstino-Pearson testing prior to Kruskal-Wallis testing. We agree with the alternative explanation that you suggest and have added additional emphasis on this. PAS-D does correlate poorly in early fibrosis because there are people with high and low PAS-D who have early fibrosis. However, the high accumulators are of interest to identify with our methods because we suspect that over time they have risk of worsening fibrosis. Our forthcoming longitudinal studies will help clarify this.

12. This is an appropriate criticism and we have edited our discussion to reflect these comments.

13. We analyzed slides to find people at the upper aspects of AAT accumulation regardless of fibrosis stage. Figure 1E is how we identified this group. We then use these groupings throughout the rest of the paper to make further comparisons with people who have high or low accumulation.

14. We used our tools to estimate polymer area in the whole population minus patients with limited tissues or tissue that wouldn’t be enough for a pathologist to make an appropriate reading. Figure 3 shows this point. Total IHC does not have a relationship with fibrosis but polymer does.

Minor Comments:

We have made the minor changes as requested.

In this manuscipt we present open source tools developed out of a large cohort of primary biopsy specimens from a rare disease population. We believe that they offer insight into the heterogeneity of disease and cellular damage in early liver fibrosis. We are grateful for the opportunity to share our work with the scientific community through PLOS ONE. We appreciate our reviewers thoughful questions and suggested revisions and have addressed them as above. Please do not hesitate to contact us with any further questions.

---

## [Decision Letter · Decision Letter 1]

14 Jun 2021

PONE-D-21-03416R1

Quantitative measurement of the histological features of Alpha-1 Antitrypsin Deficiency-associated liver disease in biopsy specimens

PLOS ONE

Dear Dr. Marek,

Thank you for submitting your manuscript to PLOS ONE. After careful consideration, we feel that it has merit but does not fully meet PLOS ONE’s publication criteria as it currently stands. Therefore, we invite you to submit a revised version of the manuscript that addresses the points raised during the review process.

As you can see, one of the reviewers still raised substantial concerns about the interpretation of your data (i.e. a clear statement which parameters show a statistically significant association with fibrosis stage a which ones do not).

We look forward to receiving your revised manuscript.

Kind regards,

Pavel Strnad

Academic Editor

PLOS ONE

Reviewers' comments:

Reviewer's Responses to Questions

**Comments to the Author**

1. If the authors have adequately addressed your comments raised in a previous round of review and you feel that this manuscript is now acceptable for publication, you may indicate that here to bypass the “Comments to the Author” section, enter your conflict of interest statement in the “Confidential to Editor” section, and submit your "Accept" recommendation.

Reviewer #1: All comments have been addressed

Reviewer #2: (No Response)

2. Is the manuscript technically sound, and do the data support the conclusions?

Reviewer #1: Yes

Reviewer #2: Partly

3. Has the statistical analysis been performed appropriately and rigorously? 

Reviewer #1: Yes

Reviewer #2: No

4. Have the authors made all data underlying the findings in their manuscript fully available?

Reviewer #1: Yes

Reviewer #2: Yes

5. Is the manuscript presented in an intelligible fashion and written in standard English?

Reviewer #1: Yes

Reviewer #2: Yes

6. Review Comments to the Author

Reviewer #1: In their work the authors performed quantitative measurement of the histological features of alpha-1 antitrypsin deficiency in biopsy specimens and stated that the quantification of polymer may identify individuals at risk for progressive disease and candidates for therapeutic interventions.

As mentioned before, the presented data are of relevance, but nevertheless presented some weaknesses in their first manuscript draft. In the revised version of their work, the authors responded to the multiple comments and edited their discussion. They emphasized again the advantage of using a standard stain for their method that is widely available across many institutions. And as such it provides valuable results.

Reviewer #2: The manuscript by Marek et al. is improved. There are still some pretty important things that need correction before this is published.

Major

1) The comparisons between fibrosis stage and IHC measurements are not statistically significant. Therefore, how can you say that both polymer and inclusions are both associated with fibrosis?

These sentences should be removed from the abstract and from the discussion (Lines 104, 443).

2) Subgroup Kruskal-Wallis testing with Dunn’s multiple comparisons should not be done (Fig 2) unless the primary analysis is significant. Therefore, the “trend” toward association between polymer and fibrosis can be mentioned. More importantly, you can discuss the reason why you were not able to prove this association (unmeasurable biopsies that limited sample number, etc).

3) There is an opportunity, even with small patient numbers to create a table of results that might make your trend lines more believable. For instance, if your message is that there is a subset of PAD-D heavy accumulators that have more fibrosis, then define a threshold at which you make your division and show a table of age, race, sex, BMI, PAS-D scores, IHC scores, and fibrosis scoring with statistics comparing the 2 columns of data. If there are other things beside PAS-D stains and fibrosis that are different, then a multivariable analysis is indicated.

Minor

1) Table 1 FEV1/FVC is misspelled

2) Discussion- line 415. Please use English not parentheses to describe the alternative of PAS-D standardization and lack of validation.

3) Fig 4D has a Y axis that is flipped

7. PLOS authors have the option to publish the peer review history of their article (what does this mean?). If published, this will include your full peer review and any attached files.

Reviewer #1: No

Reviewer #2: No

---

## [Author Response · Author response to Decision Letter 1]

29 Jul 2021

Dear Dr. Strnad and Review Committee,

Thank you for reviewing our work and offering us the opportunity to incorporate the comments and revisions requested by the review committee. Our paper entitled Quantitative measurement of the histological features of alpha-1 antitrypsin deficiency-associated liver disease in biopsy specimens uses whole slide imaging to study baseline histologic data from a large longitudinal cohort of people who have PiZZ alpha-1 antitrypsin deficiency. We demonstrate and provide new tools which can be used to estimate severity of alpha 1 antitrypsin accumulation in standard pathology specimens. We have addressed each reviewers major and minor revisions as below. We have also made editorial revisions as noted.

Editorial Revisions

Reviewer #1

Reviewer #1 did not have further requests for revision.

Reviewer #2

Reviewer #2 had several major revisions and comments. 

Reviewer #2: 

1) The comparisons between fibrosis stage and IHC measurements are not statistically significant. Therefore, how can you say that both polymer and inclusions are both associated with fibrosis?

These sentences should be removed from the abstract and from the discussion (Lines 104, 443).

2) Subgroup Kruskal-Wallis testing with Dunn’s multiple comparisons should not be done (Fig 2) unless the primary analysis is significant. Therefore, the “trend” toward association between polymer and fibrosis can be mentioned. More importantly, you can discuss the reason why you were not able to prove this association (unmeasurable biopsies that limited sample number, etc).

We performed Dunn’s multiple comparisons test after the primary ANOVA showed differences in the median PAS-D accumulation (94) samples including cirrhosis samples and Polymer IHC measurements (84 samples). Both of these analysis showed significant differences and Post-Hoc analysis was performed at this point. I have attached a charts for these below, and the results are described in the manuscript, I added additional clarification of this in our results section. An important point that we described is that when you stain a biopsy with total AAT immunohistochemistry such that polymeric and non-polymeric AAT is stained and then measured, this is not statistically significant. 

PAS-D Globule Measurements (Figure 2)

Table Analyzed PAS-D Area

Kruskal-Wallis test 

P value <0.0001

Exact or approximate P value? Approximate

P value summary ****

Do the medians vary signif. (P < 0.05)? Yes

Number of groups 5

Kruskal-Wallis statistic 27.64

Data summary 

Number of treatments (columns) 5

Number of values (total) 94

Polymeric AAT Immunohistochemistry (Figure 3)

Table Analyzed 2C1 Area

Kruskal-Wallis test 

P value 0.0002

Exact or approximate P value? Approximate

P value summary ***

Do the medians vary signif. (P < 0.05)? Yes

Number of groups 4

Kruskal-Wallis statistic 19.29

Data summary 

Number of treatments (columns) 4

Number of values (total) 84

3) There is an opportunity, even with small patient numbers to create a table of results that might make your trend lines more believable. For instance, if your message is that there is a subset of PAD-D heavy accumulators that have more fibrosis, then define a threshold at which you make your division and show a table of age, race, sex, BMI, PAS-D scores, IHC scores, and fibrosis scoring with statistics comparing the 2 columns of data. If there are other things beside PAS-D stains and fibrosis that are different, then a multivariable analysis is indicated.

This is a really good point and has been a big part of our research with this cohort already. We answered some of these questions when we performed multivariate analysis of AAT Accumulation in Clark et al. 2019 where we defined the clinical changes which are associated with higher accumulators on a qualitative scoring basis by our pathologists. This work was born out of that understanding. In figure 1C we compare these qualitative scores with our quantiative data. The “threshold” you speak of is something that is yet to be defined. The closest we are able to define in this group are outliers based on the statistical distribution of the data but it is unknown whether or not this will have lonigtudinal relevance. Our plan is to define this threshhold based on receiver-operator analysis on progression data as the cohort finishes follow-up such that we know who has developed worstening fibrosis and use these baseline data to determine a clinically relevant amount of accumulation that has sensitivity and specificity to predicting progression. 

Minor

1) Fixed

2) Fixed 

3) Fixed

---

## [Decision Letter · Decision Letter 2]

2 Aug 2021

Quantitative measurement of the histological features of Alpha-1 Antitrypsin Deficiency-associated liver disease in biopsy specimens

PONE-D-21-03416R2

Dear Dr. Marek,

We’re pleased to inform you that your manuscript has been judged scientifically suitable for publication and will be formally accepted for publication once it meets all outstanding technical requirements.

Kind regards,

Pavel Strnad

Academic Editor

PLOS ONE

Additional Editor Comments (optional):

Reviewers' comments:

Reviewer's Responses to Questions

**Comments to the Author**

1. If the authors have adequately addressed your comments raised in a previous round of review and you feel that this manuscript is now acceptable for publication, you may indicate that here to bypass the “Comments to the Author” section, enter your conflict of interest statement in the “Confidential to Editor” section, and submit your "Accept" recommendation.

Reviewer #2: All comments have been addressed

2. Is the manuscript technically sound, and do the data support the conclusions?

Reviewer #2: (No Response)

3. Has the statistical analysis been performed appropriately and rigorously? 

Reviewer #2: (No Response)

4. Have the authors made all data underlying the findings in their manuscript fully available?

Reviewer #2: (No Response)

5. Is the manuscript presented in an intelligible fashion and written in standard English?

Reviewer #2: (No Response)

6. Review Comments to the Author

Reviewer #2: (No Response)

7. PLOS authors have the option to publish the peer review history of their article (what does this mean?). If published, this will include your full peer review and any attached files.

Reviewer #2: No

---

## [Editor Report · Acceptance letter]

6 Aug 2021

PONE-D-21-03416R2 

Quantitative measurement of the histological features of alpha-1 antitrypsin deficiency-associated liver disease in biopsy specimens 

Dear Dr. Marek:

I'm pleased to inform you that your manuscript has been deemed suitable for publication in PLOS ONE. Congratulations! Your manuscript is now with our production department. 

Kind regards, 

on behalf of

Dr. Pavel Strnad 

Academic Editor

PLOS ONE